# Reusable rule-based cell cycle model explains compartment-resolved dynamics of 16 observables in RPE-1 cells

**Paul F. Lang** [1¤]*, **David R. Penas**[2], **Julio R. Banga**[2], **Daniel Weindl**[3], **Bela Novak**[1]*

**1** Department of Biochemistry, University of Oxford, Oxford, United Kingdom, **2** Computational Biology Lab, MBG-CSIC (Spanish National Research Council), Pontevedra, Spain, **3** Computational Health Center, Helmholtz Zentrum München Deutsches Forschungszentrum für Gesundheit und Umwelt (GmbH), Neuherberg, Germany

¤ Current address: Biosim, Boston, Massachusetts, United States of America
* plang@biosim.ai (PFL); bela.novak@bioch.ox.ac.uk (BN)

**Data Availability Statement:** GitHub hyperlinks to the PEtab problems (https://github.com/paulflang/cell_cycle_petab/tree/d562d3d4d19921ff851c06d0854b8b3f05c63d85), the results of the time

## Abstract

The mammalian cell cycle is regulated by a well-studied but complex biochemical reaction system. Computational models provide a particularly systematic and systemic description of the mechanisms governing mammalian cell cycle control. By combining both state-of-the-art multiplexed experimental methods and powerful computational tools, this work aims at improving on these models along four dimensions: model structure, validation data, validation methodology and model reusability.

We developed a comprehensive model structure of the full cell cycle that qualitatively explains the behaviour of human retinal pigment epithelial-1 cells. To estimate the model parameters, time courses of eight cell cycle regulators in two compartments were reconstructed from single cell snapshot measurements. After optimisation with a parallel global optimisation metaheuristic we obtained excellent agreements between simulations and measurements. The PEtab specification of the optimisation problem facilitates reuse of model, data and/or optimisation results.

Future perturbation experiments will improve parameter identifiability and allow for testing model predictive power. Such a predictive model may aid in drug discovery for cell cycle-related disorders.

## Author summary

While there are numerous cell cycle models in the literature, mammalian cell cycle models typically suffer from four limitations. Firstly, the descriptions of biological mechanisms are often inefficiently complicated yet insufficiently comprehensive and detailed. Secondly, there is a lack of experimental data to validate the model. Thirdly, inadequate parameter estimation procedures are used. Lastly, there is no standardized description of the model and/or optimization problem.

course reconstruction (https://github.com/paulflang/cell_cycle_time_course/tree/e2d373d15bea86565d38b70dcc207564edd08d6b) and the presented model versions (https://github.com/paulflang/cell_cycle_model/tree/b0a820288bc98d62333acf8434174b6446df0574) are provided throughout the text. saCeSS can be accessed via https://bitbucket.org/DavidPenas/sacess-library/branch/sacess_cell_cycle_petab. Additionally, we generated DOIs on ZENODO for the PEtab problems (https://doi.org/10.5281/zenodo.7894097), the results of the time course reconstruction (https://doi.org/10.5281/zenodo.7700130) and the presented model versions (https://doi.org/10.5281/zenodo.7700177).

**Funding:** JRB and DRP acknowledge support from grant PID2020-117271RB-C22 (BIODYNAMICS) funded by MCIN/AEI/10.13039/01100011033. The funders had no role in study design, data collection and analysis, decision to publish, or preparation of the manuscript.

**Competing interests:** The authors have declared that no competing interest exists.

To overcome these limitations, we combine best-in-class technology to address all four simultaneously. We use a rule-based model description to provide a concise and less error-prone representation of complex biology. By applying trajectory reconstruction algorithms to existing data from highly multiplexed immunofluorescence measurements, we obtained a rich dataset for model validation. Using a parallel global metaheuristic for parameter estimation allowed us to bring simulations and data in very good agreement. To maximize reproducibility and reusability of our work, the results are available in three popular formats: BioNetGen, SBML, and PEtab.

Our model is generalizable to many healthy and transformed cell types. The PEtab specification of the optimization problem makes it straightforward to re-optimize the parameters for other cell lines. This may guide hypotheses on cell type-specific regulation of the cell cycle, potentially with clinical relevance.

## Introduction

Scientific research can be seen as a collaborative process of continuously refining models of the world. Often, these refinements are driven by acquisition of new types of data, more accurate data, better data analysis methods or novel ways to describe, communicate, explain or interpret data and knowledge. In case of systems biology, we now have access to highly multiplexed measurements of biopolymer composition at single cell resolution [1–3], methods to analyse [4–11], and formats to communicate and interpret these data [12–15]. However, these methods and tools have not yet been fully exploited to refine existing models of the mammalian cell cycle control system. Through an intricate interplay of protein synthesis, degradation, phosphorylation and complexation, this cell cycle control system (1) ensures strictly alternating replication and segregation of DNA, (2) coordinates these processes with doubling of all other cellular components, (3) checks for correct completion of critical steps and (4) remains functional under a variety of perturbations [16]. Dysregulation of the cell cycle control system is associated with multiple diseases, such as cancer and neurodegenerative disorders. [17]. The complex nature of cell cycle regulation with multiple interconnected and nested feedback loops combined with the known challenges of fitting the parameters of oscillatory systems to experimental data [18] complicate the development of mechanistically detailed, yet comprehensive cell cycle models. Nevertheless, we here sought to leverage recent technological advancements to improve on existing mammalian cell cycle models, especially with regard to four dimensions: model structure, validation data, validation methodology and model reusability. We approached the challenges of this endeavour by breaking down the problem in achievable subproblems. First, we developed models of individual cell cycle transitions. Second, we fused the cell cycle transition submodels to a model of the full cell cycle. Third, we reconstructed time courses of cell cycle regulators from highly multiplexed single cell measurement. Finally we cast this data and the cell cycle model into an optimisation problem and used a powerful metaheuristic to fit the model to the data. In this manner we aggregated existing knowledge about mammalian cell cycle dynamics to an executable model that explains the dynamics of 16 observables in human retinal pigment epithelial-1 (RPE-1) cells. By combining abstract model definition in the BioNetGen (BNG) language [14, 19] with an intuitive naming convention using Human Genome Organization Gene Nomenclature Committee (HGNC) short names, and by concretely describing the model, data and optimisation problem in the PEtab format [15], we released the model in an easily contributable form to GitHub (https://github.com/paulflang/cell_cycle_petab/).

## Results

### The restriction point submodel

We start with a description of our cell cycle model, which consists of submodels for the restriction point (RP), the G1/S transition, the G2/M transition and the metaphase/anaphase (M/A) transition. All reactions (except for nuclear envelope breakdown) are modelled with mass action kinetics. For the precise meaning of abbreviations for modelled molecular species, please refer to Table A in S1 Appendix. The RP is a checkpoint in G1 that prevents cell cycle progression, until it is lifted by continuous or pulse like mitogen signalling [20], which stimulates cyclin D expression. The RP is defined as a point after which progression through the cell cycle no longer requires mitogen signalling [21]. Here, we use cyclin D:Cdk4/6 complex (CycD) as a proxy for mitogen signalling. The molecular basis of the restriction point is described by the Rb/E2f pathway shown in Fig 1A [22, 23]. E2f is a transcription factor that drives the transcription of cyclin E. Before the restriction point, E2f is kept inactive by binding to Rb [24]. However, this binding is inhibited by phosphorylation of Rb, which is mediated, for example, by CycD and CycE. The ordinary differential equations (ODEs) describing the Rb/E2F pathway are listed in Equations A in S1 Appendix. Even without E2f autoactivation (i.e. replacing the ODE for total E2f with $tE2f = 0.5$; Figs A-a and A-b in S1 Appendix) the mutual inhibition of CycE and Rb, combined with the inhibitor ultrasensitivity conferred by Rb allows for bistability in the Rb/E2f pathway. Transcriptional autoactivation of E2f (grey arrows in Fig 1A) further increases the bistable range, and thus, the robustness of this network with respect to variations in CycD concentration (Fig 1B). In agreement with the notion that once the restriction point is passed, cell cycle progression becomes independent of mitogen signalling [21], flipping the toggle switch to the high CycE state is a truly irreversible process in our submodel. Reverting the system would require negative CycD concentrations. The time course simulated with CycD just above the upper bifurcation point shows how Rb, CycE and E2f approach the high CycE steady state (Fig A-c in S1 Appendix). The increasing concentrations in CycE and E2f will trigger CycA accumulation by switching the G1/S toggle. Their values in the high CycE steady state will therefore set upper boundaries for the upper bifurcation point in the G1/S submodel.

### G1/S transition submodel

Like cyclin E, transcription of cyclin A is also activated by E2f. However, there is a delay between the accumulation of CycE and CycA proteins that shall be captured by the G1/S transition submodel. In contrast to cyclin E, cyclin A is susceptible to polyubiquitination by the Apc:Cdh1 complex. The high Apc:Cdh1 levels observed in G1 keep the CycA concentration relatively low, while the CycE concentration already rises. However, CycE and CycA can phosphorylate Cdh1, thereby preventing its association with Apc [26, 27]. Additionally, E2f also activates the transcription of Emi1 [28], which is a very slowly polyubiquitinated pseudosubstrate of Apc:Cdh1. Tight binding of Emi1 to Apc:Cdh1 further inhibits Apc:Cdh1 [29] (Fig 1C). This ODEs for this G1/S transition network are described in Equations B in S1 Appendix. As total Apc appears to be in excess of total Cdh1 and total Cdc20 [30], we assume in this submodel that all Cdh1 is bound to Apc. Similar G1/S transition submodels have been proposed by Barr et al. [31] and Novak and Tyson [32]. The mutual inhibition between CycA and Apc:Cdh1, combined with the inhibitor ultrasensitivity conferred by Emi1 allows for bistability in this G1/S transition network. This is illustrated in Fig 1C, using E2f as bifurcation parameter. Setting E2f to 0.7 $AU_{E2F}$ (i.e. just below its upper steady state in the RP submodel) and using CycE as bifurcation parameter instead, Fig B-b in S1 Appendix shows that CycA can robustly

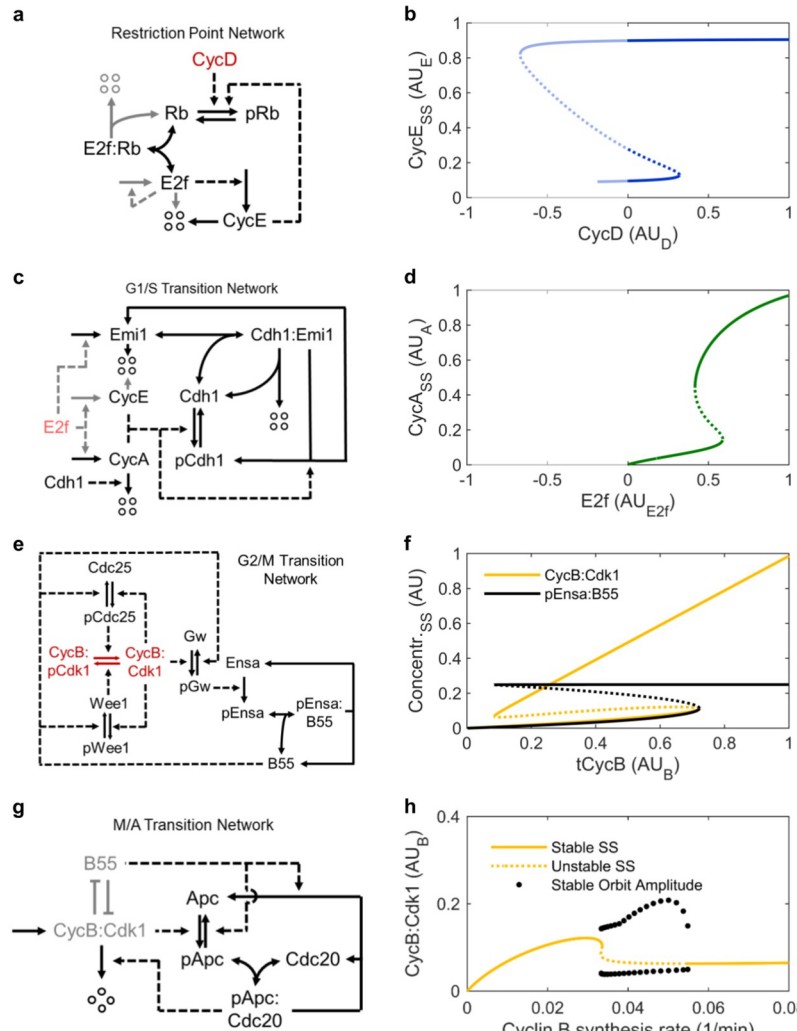

**Fig 1. Models of individual cell cycle transitions. a**, **b** Reaction network and bifurcation diagram of the restriction point. Lighter colour indicates reactions omitted in Figs A-a and A-b in S1 Appendix. **c, d** Reaction network and bifurcation diagram of the G1/S transition. Lighter colour indicates reactions omitted in Fig B-b in S1 Appendix. **e, f** Reaction network and bifurcation diagram of the G2/M transition as developed by Vinod and Novak [25]. tCycB: total cyclin B. **g**, **h** Reaction network and bifurcation diagram of the combined G2/M and M/A transition. Lighter colour represents a simplified schematic of the G2/M transition network shown in (e). In the reaction network diagrams conversions are represented by full arrows and catalytic interactions by dashed arrows. Four circles indicate degraded proteins. Red letters represent the species used as bifurcation parameters. In the bifurcation diagrams solid lines show stable and dotted lines unstable steady states. Unphysiological regions are semi-transparent. Line endings within the axes limits indicate disappearance of a steady state. For variable abbreviations please refer to Table A in S1 Appendix. Cdh1 and Cdh1:Emi in the G1/S transition network correspond to (p)Apc:Cdh1 and (p)Apc:Cdh1:Emi1, respectively in the full cell cycle model. Models incl. parameters are available in the /versions/v0.0.1/ directory of the cell_cycle_model GitHub repository.

maintain its high steady state independently of CycE. This is a critical feature of the G1/S toggle switch, since the rising concentration of CycA after flipping the switch will lead to CycE phosphorylation, marking it for SCF mediated polyubiquitination. Rising CycA concentrations will facilitate cyclin B synthesis when merging the G1/S toggle with the G2/M toggle.

## G2/M transition submodel

Cyclin B associates with Cdk1, which is kept inactive via Wee1 kinase mediated phosphorylation. Wee1 activity is counteracted by the phosphatase Cdc25. As Cdc25 is activated and Wee1 inactivated by CycB:Cdk1 (i.e cyclin B in complex with unphosphorylated/active Cdk1), this kinase phosphatase system contains two positive feedback loops. Additionally, there is mutual inhibition of CycB:Cdk1 and the phosphatase PP2A:B55 via the PP2A:B55/Ensa/Greatwall pathway, and autoactivation of nuclear CycB, leading to multiple interlinked bistable switches at the G2/M transition [33–39]. Here, we will use a model by Vinod and Novak [25], that describes the G2/M transition with a single bistable switch. Like the models for the previous cell cycle transitions, this model relies purely on mass action kinetics. According to this model, the inhibitor ultrasensitivity conferred by phosphorylated Ensa does not allow for bistability if only combined with the mutual inhibition between B55 and phosphorylated Ensa. Yet, robust bistable behaviour can be observed if this inhibitor ultrasensitivity is combined with the mutual inhibition between B55 and pGw, and/or B55 and CycB:Cdk1 (Fig 1F and Fig C in S1 Appendix). One critical property of the network is that the bifurcation parameter tCycB (i.e. cyclin B in complex with active/unphosphorylated or inactive/phosphorylated Cdk1) is an upper boundary of the system variable CycB:Cdk1 but not of B55. When reducing tCycB below the upper bifurcation point, B55 remains bound to phosphorylated Ensa, while CycB:Cdk1 almost linearly decreases with tCycB. As B55 inactivity can stabilise its inactive state almost independently of tCycB, it is possible to reduce tCycB to very low levels without activating B55. This feature will be exploited to reset the high tCycB at the G2/M transition to basal levels via the negative feedback loop represented by the M/A transition submodel.

## M/A transition submodel

The major hallmarks of the M/A transition are chromosome segregation and cyclin B degradation. Both processes are mediated by pApc:Cdc20, which is kept inactive via association to mitotic checkpoint proteins until all chromosomes are correctly attached to the microtubule spindle apparatus. The phosphorylation of Apc requires a shift in the ratio between Apc kinase and phosphatase activity. While it is well established that CycB:Cdk1 serves as Apc kinase [40], this mechanism cannot be exploited to degrade cyclin B back to baseline levels, if combined with the G2/M transition submodel and its parametrization from above. This is because CycB:Cdk1 at the upper bifurcation point is higher than CycB:Cdk1 at the lower bifurcation point (0.110 vs. 0.068; Fig 1F). Due to a negative feedback between pApc:Cdc20 and tCycB, CycB:Cdk1 would either become too low to promote further cyclin B degradation when moving down the upper branch of the steady state, or CycB:Cdk1 would become too high to allow further tCycB accumulation when moving up the lower branch of the steady state. This problem could theoretically be solved by introducing a spindle assembly checkpoint that keeps pApc:Cdc20 inactive during the process of cyclin B:Cdk1 activation/dephosphorylation, thus decoupling cyclin B:Cdk1 activation/dephosphorylation from cyclin B degradation. However, introducing stochastic events, like the attachment of kinetochores to opposite spindle poles was considered beyond the scope of the present model. Instead, we decided to implement phosphatase mediated regulation of the Apc phosphorylation state. While, not much is known about phosphatases acting on Apc in particular, it has been put forth that B55 is one of the phosphatases for Cdk1 substrates [41]. In contrast to CycB:Cdk1, pEnsa:B55 at the upper bifurcation point is (much) lower than the pEnsa:B55 at the lower bifurcation point (0.112 vs. 0.248; Fig 1F). This circumstance permits oscillations (Fig 1H, Fig D in S1 Appendix) when combining the negative feedback from the MA transition submodel (Equations D in S1 Appendix) with the G2/M switch (Fig 1G; Equations C in S1 Appendix.), in a similar way to what has

already been shown by Ferrell [42]. However, these oscillations disappear when cyclin B synthesis decreases below 0.03335 min$^{-1}$ (Fig 1H). This feature will be exploited in the full cell cycle model, where cyclin B synthesis will be reduced at the M/A transition to keep its concentration low in G1 phase.

### The core cell cycle model

Similar to previous work conducted by De Boeck *et al.* [43] we next fused the four submodels to a stably oscillating cell cycle model (Fig 2A and 2B) in a stepwise manner, accounting for additional species and reactions created through the fusion (Text A in S1 Appendix). To a large extent, the time courses of the modelled species qualitatively resembled the data in the literature [22, 44, 45]. To further test whether the full cell cycle model still has a restriction point after which cell cycle progression becomes independent of mitogen signalling, we paused the simulation at 610 min and 620 min, respectively to reduce CycD as proxy for mitogen availability from 1 to 0 AU$_D$ (Fig 2C and 2D). After continuing the simulation, we observed that cell cycle progression was halted if CycD was depleted at 610 min. However, consistent with the presence of a restriction point between 610 and 620 min after simulation start, the current round of the cell cycle was finished if CycD was depleted at 620 min. Finally, we wanted to test if our model can qualitatively capture observations from knockout experiments. Specifically, we tested if (a) CycE knockout cells can proliferate with elevated doubling time, as it has been demonstrated *in vivo* in mice [46, 47] and (b) CycA depletion leads to CycE accumulation without affecting doubling time [48]. Using the same parameter values as for the simulation of Fig 2A–2D, and without losing CycD dependence of continued proliferation, Fig 2E and 2F show that our model could reproduce both features.

### Implementing a DNA damage checkpoint

After minor model refinements such as moving the initial conditions on the limit cycle trajectory and rescaling time units (see Changelogs and model versions in Table B in S1 Appendix and simulation comparison in Fig E-a in S1 Appendix), we aimed at extending the model by a DNA damage checkpoint. However, due to the complexity of the model, extending the model in its ODE-based form would be a painstaking and error prone task. Therefore, we decided to work on a more abstract level and converted the ODE-based model to a rule-based model defined in the BioNetGen (BNG) language [14, 19] (Fig E-b in S1 Appendix) from which the ODEs are generated automatically. Using the BNG syntax, we further introduced a systematic naming convention of all molecular species, based on Human Genome Organization Gene Nomenclature Committee (HGNC) short names and known important phosphorylation sites. This naming convention is explained in Text B in S1 Appendix, and the mapping to the names used in the above models is shown in Table A in S1 Appendix. For instance, protein A with an unoccupied binding site for protein B, an occupied binding site C and a phosphorylated residue Res123 would be written as `A(B, C!+, Res123~p)`. The representation of the model in the BNG language facilitated introduction of a DNA damage checkpoint through inhibition of cyclins via binding of unphosphorylated cyclin kinase inhibitor CDKN1A (also known as p21). CDKN1A expression is controlled via TP53, a transcription factor that is activated by DNA damage. Polyubiquitination by the SKP2-containing SCF complex marks CDKN1A for degradation. Noteworthy, the mutual inhibition of CDKN1A and Cdks, combined with the ultrasensitivity conferring stoichiometric binding, results in bistability of the DNA damage checkpoint [49]. After adding CDKN1A, TP53, SKP2 and the rules coarsely described in words above, TP53 provides an interface for activating a DNA damage response checkpoint (version 3.1.0). Fig 3 shows TP53 induced reversible activation of DNA damage checkpoints

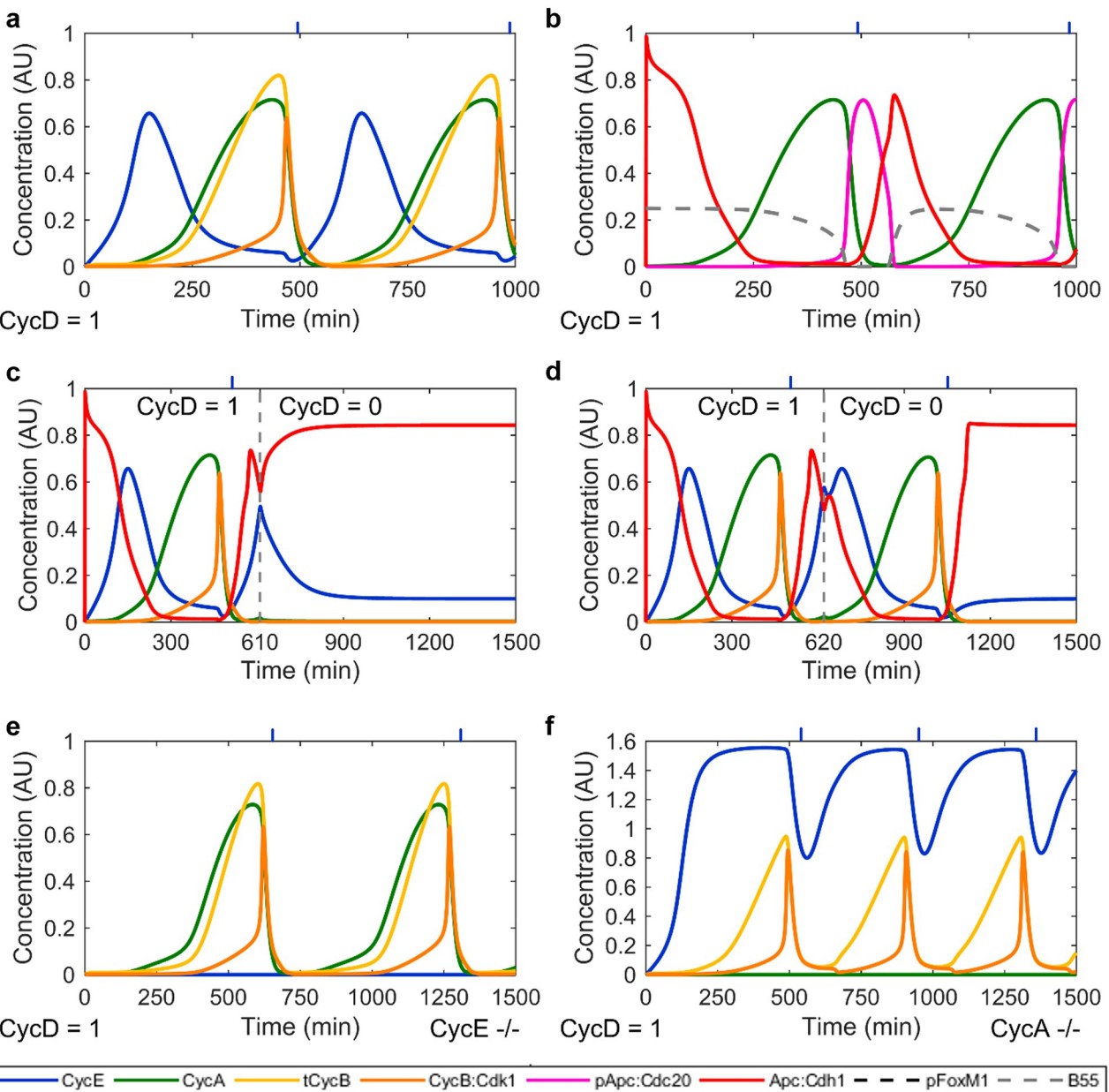

**Fig 2. Time courses of the core cell cycle model. a**, **b** Core cell cycle model at CycD = 1 AU_D. **c**, **d** Simulation of mitogen deprivation. The full cell core model was simulated as in (a, b) until 610 min (c) and 620 min (d), respectively. CycD as proxy for mitogen availability was then turned to 0 AU_D and the simulation was continued until 1500 min. **e** Full cell cycle model with CycE knockout. **f** Full cell cycle model with CycA knockout. Blue ticks at the top indicate the approximate location of the M/A transition. All models use identical parametrization (except for CycD). For variable abbreviations please refer to Table A in S1 Appendix. The model and parameters are available in the /versions/v1.0.0/ directory of the cell_cycle_model GitHub repository.

in G1 and G2 phase. Closer inspection of the G1 checkpoint indicates that CDKN1A is not fully reduced to baseline by the time the cell cycle continues, as indicated by rising CCNA/B levels. In other words, the exit point of the proliferative cell cycle trajectory at t = 20 (CCNA ≈ CCNB ≈ CDKN1A ≈ 0, CCNE ≈ 0.4) is not element of the re-entry trajectory (t between 41.667 h and approximately 55 h), as CDKN1A remains elevated for too long. We also observe

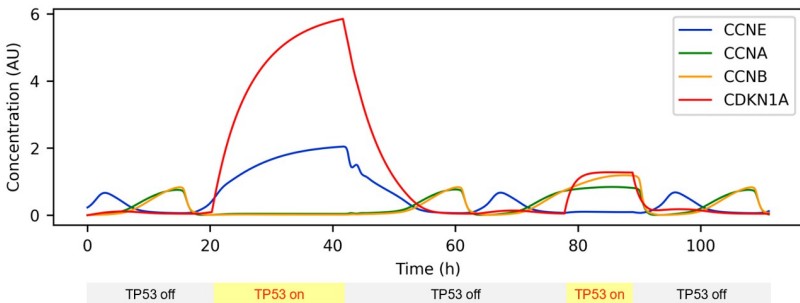

**Fig 3. Time courses with alternating TP53 levels.** DNA damage was simulated by activating TP53. The first activation corresponds to G1 phase, the second to G2 phase. Inactivation of TP53 allows continued proliferation. For better visualisation the G1 checkpoint was lifted before the steady state was reached. Model available in the /versions/ v3.1.0/cell_cycle_v3.1.0.bngl file of the cell_cycle_model GitHub repository.

high levels of the proliferative effector CCNE before CCNA/B levels rise, which may titrate some CDKN1A away from CCNA/B. These simulation results are in good agreement with Stallaert et al. [50], who found that the re-entry trajectory does not simply follow the exit trajectory in reverse direction. Instead, they argue that the re-entry trajectory follows a different path. In particular, the arrest state is not reversed through reduction of anti-proliferative effectors, but overcome through elevated expression of proliferative effectors. However, any interpretation of our simulation results must be treated with caution, as the simulation was performed with arbitrarily chosen parameters and arbitrary units of concentration. To further increase robustness of the bistability at the G1/S transition, we added CDKN1B as another stoichiometric inhibitor of cyclins, leading to version 3.2.0 (Text C in S1 Appendix).

## Introducing the notion of compartmentalisation

The model presented so far assumes uniform spatial distribution of the cell cycle regulators. However, there is substantial evidence of nucleocytoplasmic shuttling of several cell cycle regulators [51–54], including evidence that such translocation plays important roles in cell cycle regulation, for instance as a source of bistability [35]. Such nucleocytoplasmic shuttling through nuclear pores is carried out by the Ran-GTP cycle, which depends on nuclear export and import signals on the transported proteins, which can be (de)activated by posttranslational modifications [55].To enable more accurate representations of cell cycle control, we implemented the capacity for nucleocytoplasmic shuttling of cell cycle regulators into the model (Text D in S1 Appendix). This capability will later enable parameter optimisation algorithms to tune import/export rate constants. If identifiable, these rate constants may point towards translocation mechanisms of hard-to-observe species, such as complexes and certain phosphoproteins. We confirmed sustained cell cycle oscillations by simulating for more than 100 doubling times (Fig F in S1 Appendix). In addition, we implemented nuclear envelope breakdown (Fig G in S1 Appendix). However, this led to significantly reduced simulation speed. Therefore, nuclear envelope breakdown was switched off for parameter estimation.

## Obtaining multiplexed and spatially resolved time course measurements of cell cycle regulators

To estimate some of the 325 unknown parameters in the compartmental model of the cell cycle, we were searching for highly multiplexed and spatially resolved time course measurements of cell cycle regulators. We found that Stallaert and colleagues [50] published single-cell

snapshot measurements of cell cycle regulators in asynchronously dividing human retinal pigment epithelial-1 (RPE-1) cells. These measurements were obtained from iterative indirect immunofluorescence imaging (4i) experiments [56]. To reconstruct pseudo-time courses from such single-cell measurements, several cell trajectory reconstruction algorithms have been developed over the last decade [6–11]. These algorithms calculate an average/typical trajectory and rank cells according to how far they have travelled along this trajectory. For the particular case of reconstructing cell cycle trajectories, we can calculate the cell cycle time from the rank by (a) exploiting that the proliferative cell cycle trajectory forms a closed circle and (b) cell division creates two cells from one. More precisely, Kafri *et al.* [6] have shown that we can describe the probability that a randomly selected cell from a perfectly asynchronous cell population, which moves along a single circular trajectory, had its last cell division event at time *t*, with

$$p(t) = \bar{k}\left(\frac{1}{2}\right)^{\frac{t}{T}}, \tag{1}$$

where $T$ is the doubling time, and $\bar{k}$ is a scaling constant to ensure that the integral from 0 to T is 1:

$$1 = \bar{k}\int_0^T \left(\frac{1}{2}\right)^{\frac{t}{T}} dt.$$

We also know that the whole population size $N$ is the area under the curve $N \cdot p(t)$. Therefore,

$$N = k\int_0^T \left(\frac{1}{2}\right)^{\frac{t}{T}} dt,$$

and

$$k = \frac{2N\log 2}{T}.$$

Thus, the number $r$ of cells younger than $t$ is

$$r(t) = 2 \cdot N\left(1 - \left(\frac{1}{2}\right)^{\frac{t}{T}}\right).$$

Rearranging we obtain the cell cycle time from $r$ as

$$t = \frac{T}{\log 2}\log\left(\frac{r}{2N} - 1\right). \tag{2}$$

### Testing the reCAT trajectory reconstruction algorithm on simulated data

To test cell cycle trajectory reconstruction with reCAT [9], 300 cells were sampled from model version 2.1.4 (Fig 4A), such that cell density decreases exponentially over the cell cycle with one cell cycle length half-life, i.e. follow Eq (1). First, all 49 model variables were provided for trajectory reconstruction without adding noise (Fig 4B). reCAT was able to perfectly reconstruct the cell cycle trajectory from this dataset (Person correlation coefficient: $R = 1.0$; Fig 4C and 4D). As a next step, the input data was corrupted with *lognormal*$(1, 0.8^2)$ noise. Despite the substantial noise level, reCAT recovered the original cell cycle trajectory with near perfect correlation ($R = 0.997$). This accuracy remained practically unchanged ($R = 0.987$) even after

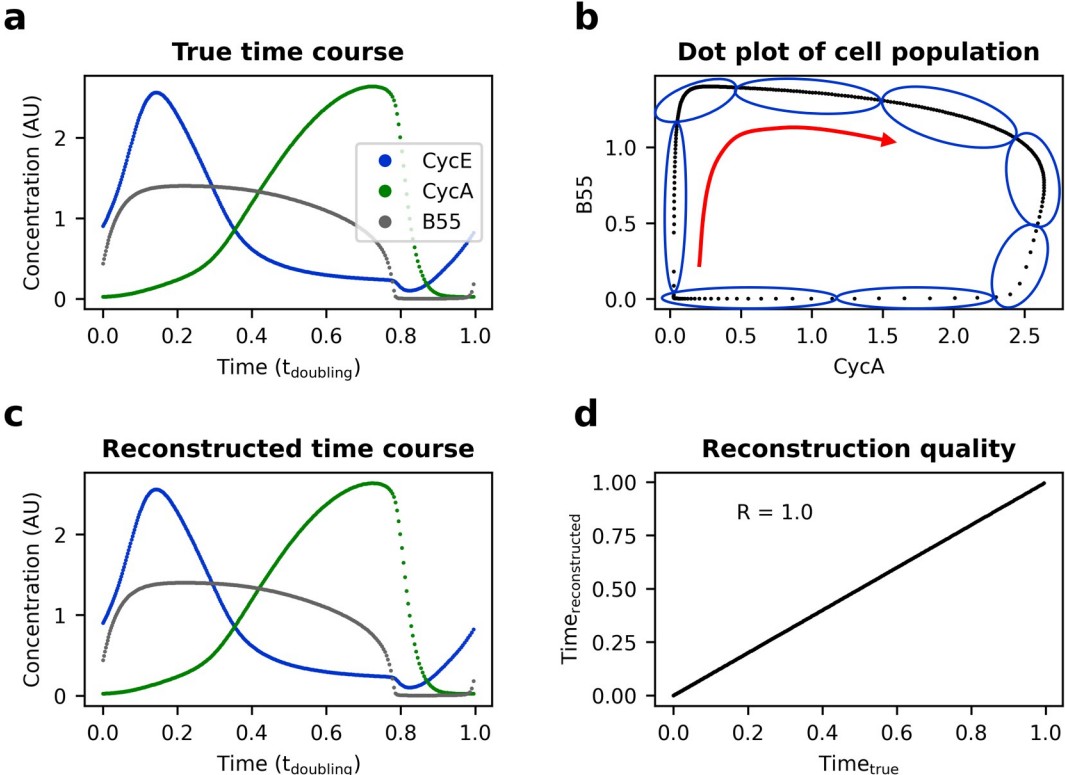

**Fig 4. Cell cycle trajectory reconstruction from noise-free simulated data with reCAT. a** 300 cells (i.e. discrete time points) were sampled across one cell cycle model simulation (version 2.1.4). Of all 49 model variables, cyclin E, cyclin A and B55 are shown. **b** Sketch of the reCAT algorithm showing the sampled cell population projected on the CycA-B55 plane. The true cell cycle time was not provided. reCAT reconstructed the trajectory by grouping the cells into 8 clusters (sketched as blue ovals) with a Gaussian mixture model and heuristically finding the shortest circular path that visits all cluster centers (sketched as red arrow). **c** Reconstructed cell cycle trajectory. **d** Correlation between true and reconstructed cell cycle time for each cell.

reducing the information content of the data by eliminating 40 variables from the reconstruction procedure (Fig H in S1 Appendix).

## Reconstructing time courses from imaging data

Stallaert *et al.* [50] obtained spatially resolved measurements of 48 cell cycle regulators (leading to a total of 292 cellular features) in thousands of RPE-1 cells. Using the dimensionality reduction method PHATE (Potential of Heat-diffusion for Affinity-based Trajectory Embedding) [57], they found that a significant proportion of RPE-1 cells exit the proliferative cell cycle trajectory into a non-proliferative G0 arm. We discarded such G0 cells prior to trajectory reconstruction (Fig I in S1 Appendix). For reasons outlined in Text E in S1 Appendix, we believe that discarding G0 cells is sufficient to safely apply Eq (2) to calculate cell cycle pseudo-times from ranks. After removal of such G0 cells, we performed trajectory reconstruction on 300 randomly selected cells (Methods). At a doubling time of 19 hours for RPE1 cells, this equates to one datapoint every 3.8 minutes on average. reCAT was only provided with the 36 of the 40 most predictive cell cycle features identified by Stallaert *et al.* that did not contain missing values. The other features were ignored during reconstruction, as they may contribute more noise than cell cycle information. Ranking the cells resulted in time courses of all 292 features, 12 of which are shown in Fig 5 (the full dataset is available in table 4i_stallaert/2021_Stallaert_

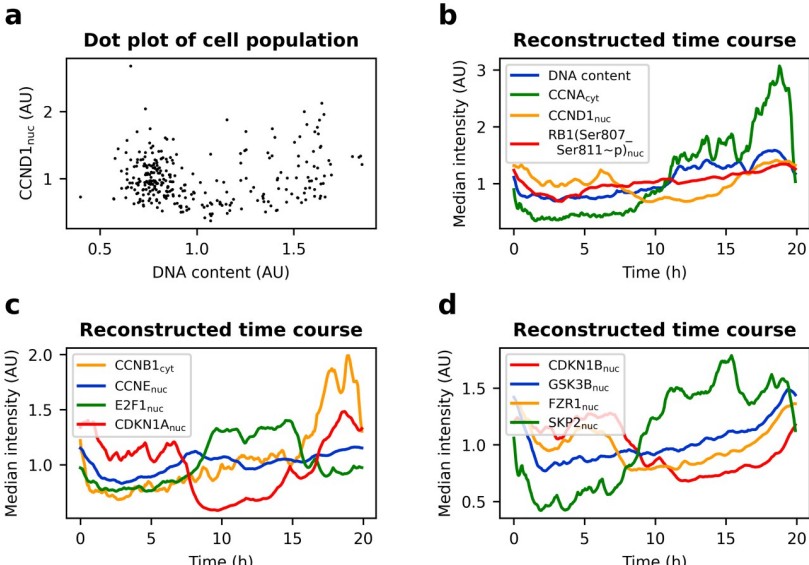

**Fig 5. Time course of cell cycle regulators in RPE-1 cells.** Proliferating cells of publicly available 4i measurements of an asynchronously dividing RPE-1 population [50] were gated in PHATE space (Fig I in S1 Appendix). For better visibility the variables are normalised to a mean of one, and axes are clipped. **a** Dot plot of 300 untreated RPE-1 cells. **b-d** Reconstructed time course of the cell population from (a). reCAT was provided with 36 variables. Time was calculated using Eq (2) and data was smoothened using a Kalman filter.

cycling_reCAT/smoothened_data.xlsx of the cell_cycle_time_course GitHub repository, and Kalman-smoothened data can be inspected with this interactive figure). Overall, the reconstructed cell cycle trajectory corresponded well with known behaviour of cell cycle regulators. For instance, CCNA accumulated in later cell cycle stages and was abruptly degraded in mitosis. CCNB followed the same pattern with a slight delay and DNA content doubled throughout the cell cycle. Interestingly however, CCNE concentration appeared more stable across the cell cycle compared to life cell imaging experiments in HeLa cells [31] and immunofluorescence measurements in U2OS cells [58].

## Parameter estimation with perfect simulated data

Having developed a compartmental model of cell cycle control and reconstructed spatially resolved time courses of cell cycle regulators, we aimed to estimate the parameters of our model. However, the optimization problem at hand involved several challenges, including: unknown parameter bounds, high dimensional parameter space, unknown parameters in the observation function that maps model states to fluorescent readout, unknown contributions to and magnitude of measurement noise, presence of multiple local optima, and structural and practical parameter unidentifiabilities. Managing these challenges is still a matter of ongoing research [18, 59–62]. To nevertheless identify an algorithm that is capable of finding the globally optimal parameters of the cell cycle model, we first performed the optimisation on idealized simulated data. These data were generated by simulating one cell cycle and sampling 100 time points for all states without introducing noise. Using multistart optimization provided with gradients obtained from adjoint-sensitivity analysis, we found 100 different and poor solutions from 100 starting points, indicating a highly rugged objective function. To avoid such premature attraction to local optima, we therefore switched to Cooperative enhanced Scatter Search (CeSS), a hybrid global-local optimisation algorithm [63] (Methods) that

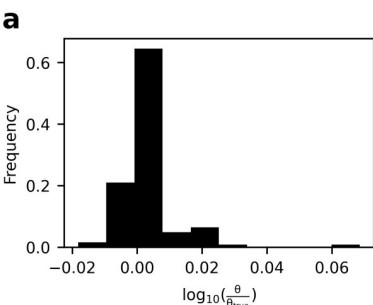
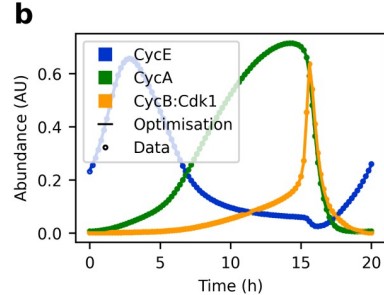

**Fig 6. Testing Cooperative enhanced Scatter Search (CeSS).** 101 evenly spaced, simulated, noise-free datapoints of 41 variables were generated with model version 2.0.0 (circles). Dynamic parameters were estimated within a $[0.1 \cdot \boldsymbol{\theta}_{true}, 10 \cdot \boldsymbol{\theta}_{true}]$ search window, where $\boldsymbol{\theta}_{true}$ denotes the ground truth parameter vector. Initial conditions were estimated within a $[eps, 1.5 \cdot \boldsymbol{\theta}_{true}]$ search window, where $eps$ denotes machine epsilon (except for Apc, where the upper boundary was 5). **a** Histogram showing distribution of all optimized parameters $\theta$ versus ground truth $\theta_{true}$, except for *Apc*: *Cdh* and *pCdh* (see text), and *kDpE2f1* and *kPhC25A* ($\theta_{true} = 0$ and $\theta \leq eps$). **b** Simulated time courses for three of the 41 variables, using the parameter vector $\boldsymbol{\theta}$ found by CeSS.

performed well in benchmark tests [64]. Optimisation took 10.7 hours using four cores of a personal computer (Intel Core i7–8550U CPU @ 1.80GHz). This time, the ground truth parameters used for generating the data were recovered with only minor deviations, except for the initial conditions of Apc:Cdh and pCdh (Fig 6A). In the cell cycle model total Cdh is constant. CeSS has recovered the correct amount for *Apc*: $Cdh(t = 0) + pCdh(t = 0) = 0.87$ up to the second decimal digit, but estimated the initial condition of Apc:Cdh higher and pCdh lower than their respective ground truth values. As the observation function was simply the pure species abundance, this confusion of initial values increases the objective function value. Therefore, the otherwise near perfect recovery of ground truth parameters indicates structural identifiability of the posed problem. Had there been structural unidentifiabilities, that is perfectly flat valleys in the objective function, it would be a highly unlikely coincidence that CeSS returns the ground truth parameters instead of any other point in the valley. Fig 6B shows the practically perfect overlap between the time course of the noise-free simulated datapoints and the simulation with estimated parameters.

## Imitating real imaging data and speeding up optimisation

Having found CeSS as an optimizer that is capable of solving the cell cycle optimisation problem on idealized data using a personal computer, the next step was to speed up optimisation to meet the demands of realistic data. To this end we used self-adaptive cooperative enhanced scatter search (saCeSS). saCeSS improves on CeSS by successive adaptation of hyperparameters, improving the cooperation strategy and timing, and combining fine- and coarse-grained parallelisation on high performance computing hardware [65]. The version used for this work is available in the sacess_cell_cycle_petab Bitbucket repository. To test saCeSS on model version 3.0.0, the optimisation problem was updated to better mimic our experimental data. In particular, the Stallaert *et al.* dataset includes five observables that are informative for this version of the cell cycle problem: CCNE, CCNA, CCNB1, E2F1 and RB1(Ser807_ Ser811 ∼ p). The expected density of the 300 datapoints is exponentially decreasing over the cell cycle, with a half-life of one doubling time. Importantly, the observables represent antibody fluorescent intensities in arbitrary units. Since antibodies differ in their affinity to their

target observables $y$ map to model species concentration via

$$y_k = o_k + s_k \sum_{i \in K} x_i, \tag{3}$$

where the index $k$ denotes different antibodies, $o_k$ represents an offset constant, $s_k$ a scaling constant, $K$ the set of all species to which the k$^{th}$ antibody binds, and $x$ their concentration. For example, the observable for the fluorescent signal from the antibody against CCNE is the sum of the model states for free CCNE and CCNE bound to (un)phosphorylated CDKN1A and CDKN1B, times an antibody-specific scaling constant, plus an antibody-specific offset constant. The resulting optimisation problem including the species-to-observable mapping from Eq (3), observable noise, simulated data, and estimated parameters with bounds was specified in the PEtab format and can be found in the /versions/v3.0.0/PEtab_PL_v3_0_0sim/ directory of the cell_cycle_petab GitHub repository. The simulated observables were corrupted with $\mathcal{N}(1, 0.1^2)$ multiplicative noise and $o_k$ was set to zero for all $k$. Since several species are no longer directly observed and scaling constants are introduced as new parameters, the problem is no longer expected to be structurally identifiable (i.e. the global optimum is a manifold of infinitely many solutions. Therefore, no biologically meaningful conclusions can be drawn from analyzing the values of unidentifiable parameters. Instead, the goal was to find a solution that provides a good fit to the data. To this end two different local solvers were tested for the optimisation: Ipopt (via parPE and using adjoint sensitivity gradients) [62], and gradient-free dynamic hill climbing (DHC).

PEtab problems specify the objective as posterior probability. Here, uniform priors and a standard deviation for the measurement noise of 0.1 were used, rendering the problem equivalent to least-squares optimisation constrained to parameter bounds. Both solvers went under the objective function value of the ground truth parameters, indicating slight overfitting. As expected, neither solver recovered the ground truth parameters, which can be attributed to overfitting and parameter unidentifiabilities. The lowest objective function value was found with the Ipopt local solver option in saCeSS. A simulation with the corresponding parameter values and the convergence curve is shown in Fig J in S1 Appendix.

## Parameter estimation with imaging data

Next, we exchanged the artificial data with real experimental data from Stallaert *et al*. This required us to make five modifications to the optimisation strategy (Text F in S1 Appendix). 'For instance, we had to fit over two cycles to enforce oscillatory behaviour, and allow the offset parameters $o_k$ to deviate from zero. All these strategies were combined into a new problem specification, using the parameters from the oscillatory but non-fitted model as initial guess. The best performance was achieved with the DHC local solver option in saCeSS. The found solution led to simulations that are in good agreement with experimentally observed time courses. Fig K in S1 Appendix shows the convergence curve and directly compares nuclear measurements and the simulation.

The optimisation procedure was repeated for model version 3.2.0, which contains relevant variables to use three additional nuclear observables in the Stallaert [50] dataset: SKP2, CDKN1A and CDKN1B. The corresponding PEtab problem can be found in the /versions/v3.2.0/ directory of the cell_cycle_petab GitHub repository. Fig L in S1 Appendix compares the simulation results using the estimated parameters to experimental measurements.

Finally, we estimated the parameters of model version 4.0.0 using nuclear and cytoplasmic observables. Since the measurement noise in the cytoplasm was much lower than in the nucleus, we allowed observable-specific noise by adding the following noise formula to the

optimisation problem:

$$\sigma_k = \sigma_1 + \sigma_{2_k} \cdot y_k.$$

$\sigma_1$ and $\sigma_{2_k}$ were estimated as part of the optimisation problem and represent additive and observable-specific multiplicative noise, respectively. $y_k$ denotes the $k^{\text{th}}$ observable and $\sigma_k$ the standard deviation of normally-distributed measurement noise. The corresponding PEtab problem can be found in the /versions/v4.0.0/ directory of the cell_cycle_petab GitHub repository. Even for this compartmental model of the cell cycle, saCeSS found a solution that leads to simulations that are in excellent agreement with experimentally observed location and time courses of the measured cell cycle regulators (Fig 7). While this solution was initially located at an edge of the search window, shifting the search window and optimising for another 40 h did not further improve the solution.

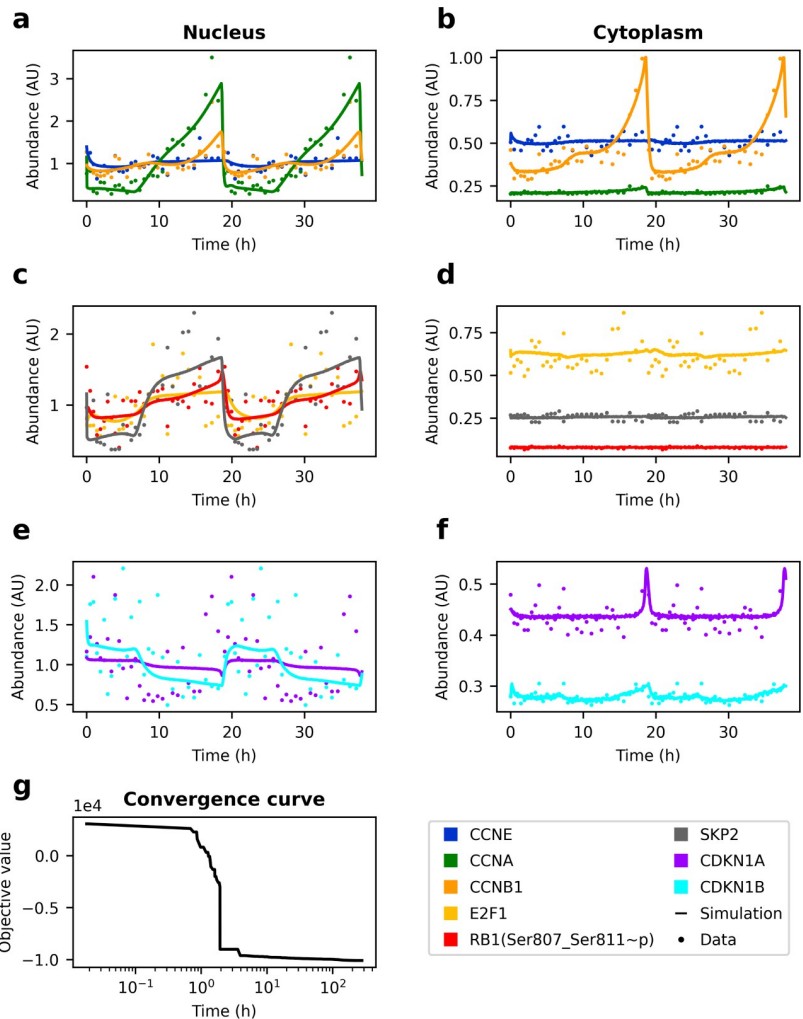

**Fig 7. Model version 4.0.0 fitted to pseudo-time courses of RPE-1 cells.** Experimental measurements and simulation results using estimated parameters. For better visibility, only every $10^{\text{th}}$ measurement is shown. **a, c, e** Nuclear compartment. **b, d, f** Cytoplasmic compartment. **g** Convergence curve. PEtab problem incl. parameter table and SBML file with optimized parameters are available in the /versions/v4.0.0/ directory of the cell_cycle_petab GitHub repository.

## Discussion

Cell cycle control is one of the most central mechanisms of life and dysregulation in mammals results in various diseases such as cancer and neurodegenerative diseases [17]. Generating and improving computational models of the cell cycle has therefore been a long-standing goal of systems biology [66–73]. With the present work, we contribute the most complete mammalian cell cycle model known to the authors that accurately explains time courses of experimentally determined observables. In particular, we improved on comparable existing models (Table 1)

**Table 1. Comparison of this work with existing cell cycle models.**

| Model | Model structure | | | | | |
|---|---|---|---|---|---|---|
| | Organisms | Checkpoints | Compartments | Species | Reactions | Parameters |
| Singhania et al. (2011) [68] | Mammals | None | CE | 4 + 6[1] | 16 | 19 |
| Gauthier et Pohl (2011) [69] | Mammals | None | CP, NU | 428 | - | 332 |
| Weis (2014) [70] | Mammals | None | CE | 25 | 73 | 136 |
| Abroudi et al. (2017) [71] | Mammals | DNA damage | CE | 66 | 138 | 150 |
| Mitra et al. (2019) [72] | Yeast | - | CE | 44 | 39 | 153 |
| Version 4.0.0 | Mammals | DNA damage | CP, NU | 123 | 630 | 294[2] |

| Model | Validation Data | | | Validation methodology | | |
|---|---|---|---|---|---|---|
| | Observables | Time points | Conditions | Objective function | Optimiser | Selection criterion |
| Singhania et al. (2011) | 2 | >10000 | 1 | - | manual | None |
| Gauthier et Pohl (2011) | 0[3] | 1 | - | - | manual | None |
| Weis (2014) | 3 | ∼ 40 | 3 | - | manual | None |
| Abroudi et al. (2017) | 0 | 0 | 0 | - | - | None |
| Mitra et al. (2019) | 10[4] | 17–24[5] | 1[6] | Least squares [7] | PyBioNetFit[8] | None |
| Version 4.0.0 | 16 | 300 | 1 | Maximum likelihood[9] | saCeSS | None |

| Model | Model reusability | | | | Remarks |
|---|---|---|---|---|---|
| | Representation | Format | Annotation | Availability | |
| Singhania et al. (2011) | ODE/predetermined logical sequence | - | - | - | - |
| Gauthier et Pohl (2011) | ODEs | Word/Matlab | None | Supplementary material | [a] |
| Weis (2014) | Reactions | SBML[10] | None | BioModels | [b] |
| Abroudi et al. (2017) | Reactions | SBML[10,11] | None | BioModels | [c] |
| Mitra et al. (2019) | Reactions, Events | SBML | None | GitHub | - |
| Version 4.0.0 | Rules or Reactions | PEtab or BNGL or SBML | HGNC | GitHub | - |

CE: cell, CP: cytoplasm, NU: nucleus.

[1] 4 ODE states, 6 Boolean states.

[2] Only estimated parameters are counted.

[3] The model was calibrated to match summary statistics about the cell and the cell cycle.

[4] mRNA instead of protein observables.

[5] Exact number differed between cell cycle synchronisation method.

[6] One condition with time course data plus 119 knock out conditions with qualitative data.

[7] Penalties for qualitative constraint violations were added to the least squares objective.

[8] Using a scatter search algorithm with a simplex local optimizer.

[9] In a strict sense, we used a maximum posterior objective, as we used priors for selected offset, scaling and noise parameters.

[10] Conversion to SBML performed by Ashley Xavier from the European Bioinformatics Institute.

[11] Simulating the SBML model with zero DNA damage does not reproduce publication results.

[a] Very detailed resolution (e.g. 12 mRNA and protein species per gene).

[b] Oscillations are not stable. Simulation of SBML model crashes after 10.8397 time units.

[c] Based on Iwamoto (2011) [75]. Oscillations are not stable.

along the four dimensions of model structure (e.g. using the notion of compartmentalisation and DNA checkpoints in the same model, all ultrasensitivity purely emerges from mass action kinetics), validation data (i.e. using the largest number of densely sampled observables), validation methodology (i.e. using a probabilistically motivated objective function and a more reproducible optimisation procedure) and reusability (i.e. providing the model and optimisation problem in multiple community standards, including an easily extensible rule-based description, tracking significant parts of model development with git version control, exploiting BNGL syntax and HGNC short names for concise semantics and releasing the model to the interactive GitHub platform). Yet, the parameters of the model were structurally unidentifiable and predictions for small molecule or genetic cell cycle perturbations could not be tested in lack of appropriate experimental time course data. Generating such data will thus be the next significant step towards improving and evaluating the model, with studies such as [74] already going into the right direction. In addition, we invite the community to contribute suggestions and improvements of any form via GitHub issues and pull requests.

## Methods

### Developing the core model

The core model and the transition models are formulated as systems of ODEs with species concentrations in arbitrary units. All parameters were chosen manually, except for *kDi-pEB*55 = 0.0068 and *kAspEB*55 = 57 in the G2/M submodel, which were experimentally determined by from Williams *et al* [76]. Criterion for the choice was to show that there exists a parametrization for which the model shows the desired behaviour (i.e. bistability for the RP, G1/S and G2M submodels, oscillations for the G2M/MA submodel, oscillations and cyclin D dependent restriction point in all unfitted model versions from `1.0.0` onwards. For `v3.1.0`, we additionally required a TP53 triggered G1 and G2 DNA damage checkpoint). The precise values are available in the cell_cycle_model GitHub repository. Nullcline plots, bifurcation diagrams and time courses where calculated with the freely available software XPP/XPPAUT [77]. For steady state bifurcation analysis the bifurcation parameter was set to zero and XPP/XPPAUT was started from the corresponding steady state. This steady state bifurcation analysis also detects Hopf bifurcations. Periodic bifurcation analysis (Fig 1H) was started from the Hopf bifurcation at $k_{SyCb} \approx 0.055$ min$^{-1}$. The biochemical reactions making up the RP, G1/S and G2M submodels were chosen to comply with experimental observations of bistability [22, 31, 33, 34, 78] in these networks. These submodels where merged in a stepwise manner to yield an RP-G1S, RP-G1S-G2M, G2M-MA and finally a full cell cycle model (RP-G1S-G2M-MA). Additional reactions and species were added whenever necessary and to make the model more comprehensive. Reaction parameters were chosen based on Williams *et al.* [76], where applicable. For the remaining parameters, an attempt was made to fulfil the following criteria: (a) Parameters shall be biologically reasonable compared to the parameters obtained by Williams *et al.* (b) Parameters in the RP, G1/S and G2M submodels shall allow for bistability. (c) After merging the submodels to the full model, parameters were adjusted to obtain sustained oscillations as observed in unperturbed cells, CycE [46] and CycA [48] knockouts to obtain model version 1.0.0.

### Including DNA damage response and compartmentalisation

Prior to interfacing with DNA damage response regulator and compartmentalisation, the model underwent rescaling, minor changes and conversion to BioNetGen Language version 2.5.2 [19] using RuleBender version 2.3.1 [79] (Table B in S1 Appendix, versions 1.0.0 − 3.0.0). The precise changes after model version 3.0.0, including the implementation

of DNA damage response and compartmentalisation were documented and tracked with Git version control and made available on the cell_cycle_model GitHub repository.

## Model verification

All relevant changes to the reaction network and all conversions between model description formats were subject to model verification. Version 1.0.0 was checked by rewriting the ODEs in a chemical reaction-based format using the freely available software COPASI [80]. The chemical reactions were automatically converted to a system of ODEs that were used for manual proofreading of the XPP/XPPAUT code. Time course simulations of XPP/XPPAUT and COPASI lead to identical results within a small margin of tolerance for numerical inaccuracy. Similarly, conversion from chemical reactions (version 2.1.4) to reaction rules (version 3.0.0) lead to identical simulation results within a small margin of tolerance. Starting with version 3.0.0, for all modifications to the reaction rules and molecule types, the expected new number of reactions generated by each rule and the expected number of total species were documented in the cell_cycle_model GitHub repository in files called `/versions/v*/sim_log_expected.log`, where the asterisk is a placeholder for the version name (for v3.0.1 `versions/v3.0.1/n_species_reactions_expected.txt` was used instead of a `.log` file). Equivalency with the auto-generated number of reactions per rule and total species (files `/cell_cycle_model/versions/v*/sim_log.log`) was confirmed by manually comparing relevant differences with the Visual Studio Code (Microsoft) file comparison tool.

## Major model assumptions

Major assumptions include: (1) All reactions (even non-elementary reactions) are best modelled with mass action kinetics (exception: Hill function as phenomenological description of nuclear pore phosphorylation by `CCNB(CDK1_Thr14_Tyr15~u)`; see Table A and Text B in S1 Appendix for the naming convention used). (2) Cyclin:Cdk complex concentration is proportional to cyclin concentration. (3) The time for protein expression is negligibly small. (4) The model environment provides all protein subunits, metabolites, components of the transcription-translation and protein degradation machinery that are required by the reactions described in the model. It also provides constant total levels of `CCND()`, `RB1()`, `FZR1()`, `ENSA_ARPP19()`, `PPP2R2B()`, `MASTL()`, `WEE1()`, `CDC25()`, `APC()`, `TP53()` and DNA (see Table A in S1 Appendix for alternative names of these proteins). (5) (De)phosphorylation of a protein is not changed by binding to another protein (exceptions: `PPP2R2B (ENSA_ARPP19)` actively dephosphorylates `ENSA_ARPP19(PPP2R2B!?, Ser62_Ser67~p)`. `FZR1(nTerm~u)` phosphorylation is in the full model assumed to be reduced due to putative steric hindrances within the

`APC(FZR!1,FBXO5!2).FZR1(APC!1,FBXO5!3,nTerm~u).FBXO5(APC!2,FZR1!3)` complex).

(6) `PPP2R2B (ENSA_ARPP19)` is a putative APC phosphatase. (7) Phosphorylated cyclin E and A are immediately degraded. (8) Effects of local enrichment of the modelled components can be ignored.

## Cleaning of experimental data

Outlier cells with respect to the following metrics were excluded from further analyses of the publicly available data from Stallaert *et al.* [81]: nuclear area, DNA content, CDKN1A (Thr145~p), cellular area, cytoplasmic DAPI stain in all rounds, and ratio between cellular and nuclear area (indicating segmentation errors). Exact thresholds are available in the curation directories of the respective datasets on the cell_cycle_time_course GitHub

repository. Values are displayed by hovering the mouse over the dashed red line indicating the threshold. Figures can be opened as described in the repository README.

### Trajectory reconstruction with reCAT

reCAT was provided with 300 cells sampled across one cell cycle model simulation (version 2.1.4). Data were transformed as described in the publication of the algorithm [9], after rescaling to a mean of $10^4$. This rescaling was closer to the order of magnitude of the input data used in the original publication and lead to excellent correlation of the reconstructed data with the ground truth. More precisely, state variables were transformed according to

$$\tilde{\boldsymbol{v}} = \log_2 \left( \frac{10^4 \cdot \boldsymbol{v}}{\frac{1}{n}\sum_{i=1}^{n} v_i} + 1 \right),$$

where $\boldsymbol{v}$ denotes a state variable and $n$ is the number of cells. reCAT was run with default settings resulting in an ordered list of cells in a cycle. The direction and start of the cycle were chosen automatically from patterns of cell cycle variables and corrected manually if necessary. For the Stallaert dataset manual correction was informed by manual annotation of cell age and phase. Raw (non-transformed) variables were divided by their mean. A Kalman filter was applied to the time series, using the R programming language with functions `StructTS` and `tsSmooth`. The time series was padded with the last 10% of the cell cycle at the beginning and the first 10% of the cell cycle in the end. `StructTS` optimised the maximum likelihood of the "level" model, where the evolution of variable $x(t)$ is described by

$$x(t+1) = x(t) + \epsilon_x, \epsilon_x \sim \mathcal{N}(0, \sigma_x^2)$$

and the measurement $y(t)$ by

$$y(t) = x(t) + \epsilon_y, \epsilon_y \sim \mathcal{N}(0, \sigma_y^2).$$

Paddings were removed and the order of the cells was converted to pseudo-time via Eq (2).

### Multistart optimisation and CeSS

Optimisation problems were specified and optimised as described by the software supplement of Villaverde *et al* [64]. For CeSS, the following hyperparameters were chosen: opts.ndiverse='auto'; opts.maxeval = 5e6; opts.log_var=[]; opts.local.solver='fmincon'; opts.local.finish='fmincon'; opts.local.bestx = 0; opts.local.tol = 1; opts.local.n1 = 100; opts.dim_refset='auto'; opts.prob_bound = 0.5; opts.n_threads = 8; opts.n_iter = 10;

   opts.maxtime_cess = 1*3600. The hyperparameters for threads 1–8 were set to dim1 = 3; bal1 = 0; n2_1 = 4; dim2 = 5; bal2 = 0.25; n2_2 = 7; dim3 = 5; bal3 = 0.25; n2_3 = 10; dim4 = 7.5; bal4 = 0.25; n2_4 = 15; dim5 = 10; bal5 = 0.5; n2_5 = 20; dim6 = 12; bal6 = 0.25; n2_6 = 50; dim7 = 15; bal7 = 0.25; n2_7 = 100; dim8 = 15; bal8 = 0.5; n2_8 = 10000000000. AMICI was used for model simulations and gradient calculations [82].

### Optimisation with saCeSS

All problems were specified in the PEtab format and can be found in the cell_cycle_petab GitHub repository. saCeSS was interfaced with parPE `v0.4.9` [62] and used AMICI `v0.11.28` for model simulations and gradient calculations (code available in the sacess_cell_cycle_petab Bitbucket repository). Default settings of saCeSS and parPE were used (saCeSS hyperparameters are automatically adapted based on the number of islands). The number of

runs (coarse-grained parallelisation) and processors (fine-grained parallelisation) is documented in /versions/PEtab_PLang_problem_v43.xlsx of the cell_cycle_petab GitHub repository.

## Data visualisation and illustrations

Data were visualised using MATLAB (R2018a) and Python (v3.9.5 and v3.10.3) with the matplotlib (v3.4.3 and v3.5.1) package. For plotting Fig F in S1 Appendix the first cell cycle was eliminated from the simulation results as its initial conditions do not match the conditions at the start of subsequent cycles. Interactive online figures were were created with the plotly library (v4.9.1) in R (v3.6.1). Reaction Networks were drawn with Microsoft PowerPoint.

## Supporting information

**S1 Appendix. Supporting tables, equations, figures and notes.** The Supporting Tables describe the variables of the core model and the changelog of model versions. The ODE systems for each cell cycle submodel are provided in Equations A-D in S1 Appendix. The Supporting Figures show additional analysis of the models, trajectory reconstruction and parameter estimation. Supporting Texts in S1 Appendix. discuss merging of submodels, naming conventions in the BioNetGen model, adding CDKN1B to the cell cycle model, introducing compartmentalisation, considerations on the effect of cell cycle arrest and trajectory reconstruction, and handling of real-world data in parameter estimation.
(PDF)

## Acknowledgments

The authors would like to thank Frank Stefan Heldt, John Sekar and Jonathan Karr for inspiring the conceptualisation of this work.

PFL received support from grant EP/L016494/1 provided by the University of Oxford and the EPSRC & BBSRC Centre for Doctoral Training in Synthetic Biology. BN acknowledges research support from the BBSRC Strategic LoLa grant BB/M00354X/1. DW acknowledges funding from the German Federal Ministry of Education and Research (BMBF) within the e:Med funding scheme (grant no. 01ZX1916A). The authors acknowledge CESGA (Centro de Supercomputació́n de Galicia) for providing access to its FinisTerrae III supercomputer.

## Disclaimer

The authors declare that the model is protected by an Academic Use License (https://github.com/paulflang/cell_cycle_petab/blob/main/LICENCE) and not for use by commercial businesses, unless licensed by Oxford University Innovation Ltd.

## Author Contributions

**Conceptualization:** Paul F. Lang.

**Data curation:** Paul F. Lang.

**Funding acquisition:** Bela Novak.

**Investigation:** Paul F. Lang, David R. Penas.

**Methodology:** Paul F. Lang, Julio R. Banga, Bela Novak.

**Project administration:** Paul F. Lang.

**Resources:** Julio R. Banga, Bela Novak.

**Software:** Paul F. Lang, David R. Penas, Daniel Weindl.

**Supervision:** Bela Novak.

**Validation:** Paul F. Lang, David R. Penas.

**Writing – original draft:** Paul F. Lang.

**Writing – review & editing:** Julio R. Banga, Daniel Weindl, Bela Novak.

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
