## [Decision Letter · Decision Letter 0]

29 Jun 2023

Dear Dr Lang,

Thank you very much for submitting your manuscript "Reusable rule-based cell cycle model explains compartment-resolved dynamics of 16 observables in RPE-1 cells" for consideration at PLOS Computational Biology.

As with all papers reviewed by the journal, your manuscript was reviewed by members of the editorial board and by several independent reviewers. In light of the reviews (below this email), we would like to invite the resubmission of a significantly-revised version that takes into account the reviewers' comments.

We cannot make any decision about publication until we have seen the revised manuscript and your response to the reviewers' comments. Your revised manuscript is also likely to be sent to reviewers for further evaluation.

Sincerely,

Jing Chen

Guest Editor

PLOS Computational Biology

Douglas Lauffenburger

%CORR_ED_EDITOR_ROLE%

PLOS Computational Biology

Reviewer's Responses to Questions

**Comments to the Authors:**

Reviewer #1: In this study, Lang et al. present a comprehensive model of cell cycle progression that successfully captures cell cycle effector dynamics obtained experimentally in RPE-1 cells using single-cell iterative immunofluorescence. This model is composed by fusing submodels of the effector dynamics governing four key cell cycle state transitions. The authors performed parameter estimation using a parallel global optimization metaheuristic to fit the model to the data. This work represents a significant improvement over previous attempts to model the cell cycle in its entirety, making excellent use of the RPE-1 dataset to reconstruct the dynamics of 48 cell cycle proteins, including changes in their compartmentalization. I believe that this work is rigorous and important and should be published after the following minor points are addressed:

The authors construct a rule-based model of the DNA damage checkpoints in G1 and G2, successfully capturing known dynamics of cyclins E, A, and B. They also note that CDKN1A does not fully return to baseline prior to cell cycle reentry, consistent with a previous report by Stallaert et al. However, in this same paper, the authors demonstrate that proliferative effectors such as cyclin D and CDK4 increase with CDKN1A during cell cycle arrest, while total RB decreases, and that these changes may allow cells to reenter the cell cycle in the presence of residual CDKN1A. Does the model also capture these dynamics when TP53 is turned off? If so, can the authors provide any additional mechanistic insight into how cell cycle reentry is governed following DNA damage?

The authors successfully incorporate nucleocytoplasmic shuttling of cell cycle proteins into their model. One important regulatory event that uses effector compartmentalization to govern state cell cycle state transitions is the spatial positive feedback at the onset of mitosis described by Santos et al. (PMID: 22726437). Does the current model also capture this mechanism? If so, this would be an excellent example of another cell cycle mechanism explained by the model, and could be mentioned in the manuscript.

The authors mention that predictions of perturbations could not be tested due to lack of experimental time course data. In a follow-up to the manuscript providing the experimental data supporting this manuscript (Stallaert et al., Cell Systems 2022), similar single-cell measurements of cell cycle effectors in response to perturbations, including DNA damage (Stallaert et al, MSB 2022, PMID: 36161508). I do not believe that modeling these new data is required in this manuscript; however, this seems like an obvious follow-up to this study to further test the model, and could be cited in the discussion when describing future steps.

Reviewer #2: Summary:

The manuscript “Reusable rule-based cell cycle model explains compartment-resolved dynamics of 16 observables in RPE-1 cells” by Paul et al. improves cell cycle models on model structure and utilizes the model to validate experimental data. The comprehensive model explains the behaviour of human cells and the model simulations are in agreement with the experimental results. This is an excellent study combining mathematical model with experiment, especially there is a significant lack of study of validation a comprehensive mathematical cell cycle model on experimental data. There are several points that should be addressed before publication.

1. In the Author summary, the authors mentioned that the descriptions of biological mechanisms are “overly complicated” yet “insufficiently comprehensive and detailed”. Could the authors explain the meaning of this? By which means the descriptions of biological mechanisms are overly complicated? And by which means the descriptions of biological mechanisms are insufficiently comprehensive and detailed? To my understanding, trying to make the cell cycle model comprehensive and detailed results in the complication.

2. In line 123 at the G2/M transition submodel part, authors wrote “As B55 inactivity can stabilize its inactive state almost independently of CycB:Cdk1,…”. In Figure 1e, CycB:Cdk1 will increase pGw, increase pEnsa, then increase pEnsa:B55. CycB:Cdk1 will cause the inactivation of B55, why B55’s inactive state is independent of CycB:Cdk1? Or B55 inactivity can stabilize its inactive state almost independently of tCycB?

3. In line 133 at the M/A transition submodel part, authors wrote “In contrast to CycB:Cdk1, pEnsa:B55 at the upper bifurcation point is much lower than the pEnsa:B55 at the lower bifurcation point (Fig. 1f), allowing to couple the bistable G2/M switch with B55 driven negative feedback (Fig. 1g)”. In Fig. 1f, what is the approximate value of pEnsa:B55 at the upper bifurcation point and at the lower bifurcation point? What is the approximate value of CycB:Cdk1 at the upper bifurcation point and at the lower bifurcation point? I saw that the difference between pEnsa:B55 is at around 0.2 between the two bifurcation points, and the difference between CycB:Cdk1 is about 0.7. What does the pEnsa:B55 “much lower” mean?

What is the meaning of “…, allowing to couple the bistable G2/M switch with B55 driven negative feedback (Fig. 1g)”? The much lower of pEnsa:B55 at the upper bifurcation point allow the limit cycle exist after coupling the bistable G2/M switch with B55 driven negative back? Could the authors give some explanations of the importance of pEnsa:B55 at the upper bifurcation point on coupling G2/M and MA? What will happen if the “much lower” doesn’t exist?

4. In line 137 at the M/A transition submodel part, authors wrote “Combining the negative feedback…. results in limit cycle oscillations…”. I think that the dCycB:Cdk1/dt equation (17) will become the dCycB:Cdk1/dt equation (26) in the supplemental file. Does this also mean that the saddle-node bifurcation in Fig. 1f of CycB:Cdk1 will become the Hopf bifurcation in Fig. 1h by change the Cyclin B synthesis rate (ksyCb in equation (26)) at the supplemental file?

5. In line 145, author wrote “Without changing parameters, we could further demonstrate that the model also agrees with knockout experiment of CycE [38, 39] or CycA [40].” As there still is a stably oscillating cell cycle, does this mean that cells can still finish the cell cycle after CycE or CycA knockout?

6. After fused the four submodels to stably oscillating cell cycle model, I believe that there are two cell cycles shown in Fig. 2a, b. It will be helpful if the authors can label the start and end of each cell cycle on the top of the plot. Is the Hopf bifurcation still exist after fusing the four submodels? Is the Hopf bifurcation in Fig. 1h still exist by changing the bifurcation parameter CyclinB synthesis rate after fusing the four submodels? If the Hopf bifurcation still exist, the bifurcation parameter CyclinB synthesis rate range and amplitude of the limit cycle will be different?

The orange line represents the CycB:Cdk1 and its peak is about 0.6 then decrease, during this 400 to 500 min, does the CycB:Cdk1 reaches a limit cycle? Because the limit cycle amplitude shown in Fig. 1h is about 0.2, I am not sure that if the Hopf bifurcation is still exist after fusing the four submodels.

7. To my understanding, the stably oscillating is most important characteristic after fusing these four submodels. There are experimental evidences showing the existence G1/S transitions and G2/M transitions. But methodologically, could the authors give some explanations on the necessity of the toggle switch of the G2/M submodel, Hopf bifurcation after combining the negative feedback of M/A transition, or the bifurcation in G1/S for the stably oscillating after fusing these four submodels?

8. In Fig. 2a,b,c,d, is the stably oscillating independent of the initial values of the state variables? For example, if CycD = 1, the ODE simulation will be stably oscillating no matter of the initial values and there is no stably oscillating no matter of the initial values if CycD = 0?

9. In line 170, could the authors give some explanations on “the re-entry trajectory does not follow the same path as the exit trajectory”? Please point out the simulations at which time periods in Fig. 3 are not the same?

10. What does the cell density p mean? Is it the density of the cell number? In line 192, could the authors give some explanation on equation (1)? Will r be calculated from the imaging data in order to calculate the t? What does r mean in the imaging data and how to calculate it?

11. In line 195, what is the meaning of “cell density decreases exponentially over the cell cycle with one cell cycle length half-life.”?

12. Duration the parameter estimation, are the initial values of the model variable treated as parameters and estimated? Or the initial values are fixed?

13. In line 258, authors wrote “…were rescaled to a mean of one.” I saw that the protein values in Fig. 6b, d don’t have a mean of one, what does “…rescaled to a mean of one’ mean?

14. In line 374, does the variable v mean a state variable in the model? Could the authors give some explanations on the equation v tilde?

15. After the parameter estimation the imaging data, these parameters can obtain sustained oscillations and fit the data. What about the bistability of the submodel? I saw the model are modified based on Supplementary Note 1: Merging submodels, if these modifications are omitted after optimization on the imaging data, will the submodels still have the bistability show in Fig. 1 based on the estimated parameters? To my understanding, during the parameter procedure, the optimization algorithm doesn’t guarantee the existence of the bistability for each submodel and the submodel could lose its bistability with the estimated parameters? If optimization procedure doesn’t guarantee the existence of the bistability in submodels, similar to my question 7, what is the necessity of the bistability in the submodels when used the full model to fit the experimental data? Are the sustained oscillations the only necessity when trying to fit the experimental data because the bistability in submodel could not be exist after the optimization?

16. There is a 0 to the power of 3 in Table 1 Observables column Gauthier et Pohl (2011). What does this mean?

17. Could the authors provide a Parameters Table similar to the Table 1 at the supplemental file?

18. In the Fig. 3 at the supplemental file, pWee1, Cdc25 and Gw are explained in the captions, but they are not shown in the plots.

19. What is the criteria used in classified G0 cells shown in Fig. 9 at the supplemental file? Are there xaxis and yaxis in the plot?

Reviewer #3: In this work, Lang and co-workers present a comprehensive framework using mathematical models to explain measured mammalian cell cycle dynamics. They focus on advancing and integrating four key aspects: defining a model structure, processing measured data, using these data to fit optimal model parameters, and ensure re-usability through standardization. They incorporate existing submodels of the cell cycle into a model for the whole cell cycle, showing regular cell cycle oscillations. Next, they discuss a way to obtain time courses of multiple cell cycle regulators using the recently published dataset for RPE-1 cells by Stallaert et al. [ref. 42]. Finally, they discuss different ways to address the parameter optimization problem starting from synthetic data and the measured data, illustrating that one can recover the ground truth with the synthetic data.

I enjoyed reading this work and really appreciated their approach, ranging from model construction, over parameter estimation and optimization problems, to processing and interpreting measured data. Such a wide spectrum of techniques is not commonly used and its integration is a main strength of this work. It presents an excellent standardized starting point to explore how mathematical models can explain measured data in the context of the cell cycle. I believe that such efforts can play an important role in better understanding the dynamic functioning of the cell cycle, and eventually predicting cell-cycle behavior (e.g. in the context of drug development and cell cycle-related disorders). Here below, I provide a list with comments and questions meant to further improve readability and interest of the work. When properly addressed, I would recommend publication in PLoS Computational Biology.

Main comments/questions:

1. Model parameters.

- I did not find any table of used model parameters? This would be good to add, certainly for the simulations shown in Figs. 1-2. As the number of parameters (e.g. mentioned in Table 1) is very large, in particular for model 4.0.0, I understand it is difficult to put all parameters here in a Table. However, perhaps the most important ones and/or the ones that are experimentally confirmed can be added (see next point)?

- How were the model parameters chosen? Some reasoning is explained in the text, but which parameters are chosen based on experimental measurements of rate constants / concentrations / etc., and which are more arbitrary? This would be good to mention (in the table), or only show the experimentally motivated parameters.

- Which parameters do you find from the optimization, both starting from the synthetic data (mimicking experimental data) and the actual experimental data? Are the estimated parameters logical, e.g. do they correspond to experimentally measured rates? Do we learn something new from the estimated parameters in terms of how the network functions dynamically?

- For the perfect synthetic data, you show that the ground truth can be recovered, so you find the global minimum. For the rescaled synthetic data and the experimental data, this is not discussed. Could you say something more on how often you find different local optima, whether those perhaps approximate the data very well too, and how they differ in their parameter profiles?

- Saying something more about the influence of sampling rate and noise amplitude on parameter estimation could be informative.

2. Model structure.

- How does this extended cell cycle model differ from other cell cycle models? The authors compare some key aspects in Table 1, but I would find it interesting to expand the discussion on the novelty of the current model vs. such previous models in the main text. For example, how does this molecularly detailed model differ from others which also aim to incorporate molecular detailed interactions? Recently, De Boeck et al. (PloS CB 17, e1009008, 2021) also introduced a cell cycle oscillation model that incorporates the various bistable switches mentioned here. However, they omitted the molecular detail underlying each bistable curve, focusing only on the functional motifs, leading to much less parameters that need to be estimated.

- (Related to point 1) If you fit the different model versions to experimental data, to which extent do key model parameters stay the same or change significantly? As the title includes "compartment-resolved", how do the model parameters change in the model version with/without compartments. If the estimated parameters change with every model version, I wonder whether this is a problem? As new interactions are also still being discovered, I can imagine that one can never really capture all molecular details, so perhaps one wants to see that a large part of the parameters remains unchanged across model versions.

3. Time-course reconstruction.

To which extent does the reconstruction of a time course from snapshots of many different cells give a realistic picture of the behavior of an individual cell? Some short discussion and visualization (sketch) on how this reconstruction works would be useful. I guess this generates an averaged cell cycle trajectory? However, I expect individual cells to follow quite different trajectories, corresponding to (strong?) heterogeneities in the equivalent model parameters describing each cell (perhaps even time-dependent parameters in response to changing conditions). Does this lead to challenges for the reconstruction of the cell cycle time courses and model fitting? Later in the paper, synthetic data is generated from a model with fixed parameters in the case with/without noise on the variables/observables. What if noise on the parameter values is introduced?

Minor comments/questions:

a. In the summary, the authors write "the descriptions of biological mechanisms are often overly complicated yet insufficiently comprehensive and detailed." It seems contradictory that the model is overly complicated, yet not detailed enough? This is repeated in the introduction, but without the "detailed", the contrast between detail/complicatedness vs comprehensive/understandable does make sense.

b. L43: "tools have not yet been exploited to refine existing cell cycle models". This statement seems a bit too strong. Researchers have used those different elements to refine models, but rather not in the same way and to the same extent as here?

c. L78 (and other places): "tE2f" -> it would improve readability to define this in the main text. This also goes for the other instances where a "total amount" is used. It would also be easier if this were written more explicitly in the figures, e.g. Fig. 1f: tCycB -> total amount/concentration of CycB.

d. L80-81: one refers to Fig. 1B here, but I do not see how this sentence relates to Fig. 1B other than that there is bistability.

e. Fig. 1b and Fig. 1d: Why are you plotting negative values on the x-axis, which is not biologically/physically relevant? In Fig. 1b I understand as you might want to show the whole curve, although I would (also graphically) indicate that the negative value range is not physical. In Fig. 1b I also find it strange how the bottom branch terminates: why there in the middle of the graph (around (-0.2,0.05))?

f. L113-115: The authors seem to limit themselves here to recent studies specifically focusing on the PP2A:B55/Ensa/Greatwall pathway in the mammalian cell cycle, although I think it could be nice to mention several other experimental works here that show the existence of multiple bistable switches in the G2/M transition: ref [70], ref [71], Mochida et al. Curr Biol 26, 3361 (2016), and Kamenz et al. Curr Biol 31, 794 (2020).

g. In Fig. 1h and Fig. S4, the authors show that there are limit cycle oscillations in the system. However, it seems that the oscillations do not "go around" the whole S-shaped curve, so it does not look like a typical relaxation-like oscillator. Why is this the case and does this also happen in the optimized models later in the manuscript? Could the authors give some interpretation? Is this because there are multiple bistable switches (if so, other projections could be informative)?

h. L148: it could be useful to shortly mention the main changes here?

i. L149-151: why is extending the ODE-based model with additional interactions so difficult? This is not immediately clear to me. Is it just because having a visual, rule-based initial layer that automatically translates interaction pictures into ODEs is easier for the user and one is less likely to make a typo in equation terms? Is it true that eventually, using this BNG interface, one goes back to a system of coupled ODEs? In this sense one is not solving a different type of model, but it is just an easier user interface to get to the eventual ODE system? The BNG system also has the option to translate it to a Gillespie problem, which is useful, but not used here I think? I would find a short discussion to clarify these points useful.

j. L158-160: How did the BNG formulation facilitate the introduction of the DNA damage checkpoint, in which sense was this not possible (or much more difficult) in the regular ODE formulation? Please explain a bit more.

k. L196-199: A figure sketch as to how this method works with intermediate steps (sampling - adding noise - reconstruction) could be useful, even in a main figure. Similar remark when talking about mapping of observables. These are all important in handling the experimental data, so having it visually clear in a sketch could be nice for the reader.

l. L211-213: One talks about 36 features (out of 40), and then about 292 features. Where does the number 292 come from?

m. Fig 5: The caption is hard to entirely understand as e.g. \\theta_{true} and \\eps are not defined/discussed in the main text.

**Have the authors made all data and (if applicable) computational code underlying the findings in their manuscript fully available?**

Reviewer #1: Yes

Reviewer #2: Yes

Reviewer #3: Yes

PLOS authors have the option to publish the peer review history of their article (what does this mean?). If published, this will include your full peer review and any attached files.

Reviewer #1: No

Reviewer #2: No

Reviewer #3: No
---

## [Decision Letter · Decision Letter 1]

30 Oct 2023

Dear Dr Lang,

Thank you very much for submitting your manuscript "Reusable rule-based cell cycle model explains compartment-resolved dynamics of 16 observables in RPE-1 cells" for consideration at PLOS Computational Biology. As with all papers reviewed by the journal, your manuscript was reviewed by members of the editorial board and by several independent reviewers. The reviewers appreciated the attention to an important topic. Based on the reviews, we are likely to accept this manuscript for publication, providing that you modify the manuscript according to the review recommendations.

Please address the few minor comments by Reviewer 2.

Sincerely,

Jing Chen

Guest Editor

PLOS Computational Biology

Pedro Mendes

Section Editor

PLOS Computational Biology

Please address the few minor comments by Reviewer 2.

Reviewer's Responses to Questions

**Comments to the Authors:**

Reviewer #1: The authors have adequately addressed all of my issues.

Reviewer #2: The authors have largely addressed the questions. I only have several minor concerns.

1. In line 140, the author wrote “introducing a spindle assembly checkpoint that keeps pApc:Cdc20 inactive during the process of cyclin B:Cdk1 activation/phosphorylation, thus decoupling cyclin B:Cdk1 activation/phosphorylation from cyclin B degradation.” Does this represent the equation (25) in Supporting Equations 4 that pApc:Cdc20 only degrades tCycB but CycB:Cdk1 complex? Is this modification biologically reasonable? It seems that Cdk1 protects CycB from the degradation caused by pApc:Cdc20.

2. A following-up question to the previous question 5, so the “continued cell proliferation” is considered as the existence of sustained oscillations of proteins, e.g., CycE, CycA and CycB:Cdk1. The other characteristics as period and amplitude of the limit cycle are not considered?

3. In addition to the Supporting Note 5, will the removal of G0 cells from an asynchronous cell population help the parameter fitting procedure? Like the question 15, as the authors said, the parameter fitting procedure only enforces behavior seen in the data for fitting. But the optimization doesn’t enforce the oscillatory behavior by itself. Is this another reason only used data for oscillatory behavior because there is no oscillatory behavior in G0 cells, and using the G0 cells will undermine the optimization process?

4. In the table 2 of the Supporting Information, there are changes of 2.1.0 and 3.0.1 about the transcription factor E2f. What is necessity using E2f:PX as the transcription factor instead of simply using the free unphosphorylated E2f as the transcription factor? Will using E2f bound to the promoter as the transcription factor increase the robustness of the RP or G1/S transition in the submodel?

5. In Supporting Note 6, what is modification to account for “simulations with these optimised parameters unexpectedly showed small dampened oscillations of observables for a short period of time.” I think that the modification would be the introduce of Ok. But in line 300, authors wrote “Ok was set to zero for all k”. Setting Ok to zero is only applied at Imitating real imaging data but not at Parameter estimation with imaging data?

Reviewer #3: Thank you for thoroughly addressing all comments/questions. This is very nice work and I am happy to recommend it for publication.

**Have the authors made all data and (if applicable) computational code underlying the findings in their manuscript fully available?**

Reviewer #1: None

Reviewer #2: Yes

Reviewer #3: Yes

PLOS authors have the option to publish the peer review history of their article (what does this mean?). If published, this will include your full peer review and any attached files.

Reviewer #1: No

Reviewer #2: No

Reviewer #3: No

Figure Files:

Data Requirements:

Reproducibility:

References:

---

## [Decision Letter · Decision Letter 2]

24 Nov 2023

Dear Dr Lang,

We are pleased to inform you that your manuscript 'Reusable rule-based cell cycle model explains compartment-resolved dynamics of 16 observables in RPE-1 cells' has been provisionally accepted for publication in PLOS Computational Biology.

Best regards,

Jing Chen

Guest Editor

PLOS Computational Biology

Pedro Mendes

Section Editor

PLOS Computational Biology

Reviewer's Responses to Questions

**Comments to the Authors:**

Reviewer #2: The authors have thoroughly addressed all my questions.

**Have the authors made all data and (if applicable) computational code underlying the findings in their manuscript fully available?**

Reviewer #2: Yes

PLOS authors have the option to publish the peer review history of their article (what does this mean?). If published, this will include your full peer review and any attached files.

Reviewer #2: No

---

## [Editor Report · Acceptance letter]

18 Dec 2023

PCOMPBIOL-D-23-00708R2 

Reusable rule-based cell cycle model explains compartment-resolved dynamics of 16 observables in RPE-1 cells

Dear Dr Lang,

I am pleased to inform you that your manuscript has been formally accepted for publication in PLOS Computational Biology. Your manuscript is now with our production department and you will be notified of the publication date in due course.

With kind regards,

Bernadett Koltai
